# Comparison of Joint Kinematics in Transition Running and Isolated Running in Elite Triathletes in Overground Conditions

**DOI:** 10.3390/s21144869

**Published:** 2021-07-16

**Authors:** Laura Fraeulin, Christian Maurer-Grubinger, Fabian Holzgreve, David A. Groneberg, Daniela Ohlendorf

**Affiliations:** Institute for Occupational Medicine, Social Medicine and Environment Medicine, Goethe University Frankfurt, Theodor-Stern-Kai 7, Building 9a, 60590 Frankfurt/Main, Germany; maurer-grubinger@med.uni-frankfurt.de (C.M.-G.); holzgreve@med.uni-frankfurt.de (F.H.); occup-med@uni-frankfurt.de (D.A.G.); ohlendorf@med.uni-frankfurt.de (D.O.)

**Keywords:** triathlon, running, statistical parametric mapping, kinematics, inertial motion capture

## Abstract

Triathletes often experience incoordination at the start of a transition run (TR); this is possibly reflected by altered joint kinematics. In this study, the first 20 steps of a run after a warm-up run (WR) and TR (following a 90 min cycling session) of 16 elite, male, long-distance triathletes (31.3 ± 5.4 years old) were compared. Measurements were executed on the competition course of the Ironman Frankfurt in Germany. Pacing and slipstream were provided by a cyclist in front of the runner. Kinematic data of the trunk and leg joints, step length, and step rate were obtained using the MVN Link inertial motion capture system by Xsens. Statistical parametric mapping was used to compare the active leg (AL) and passive leg (PL) phases of the WR and TR. In the TR, more spinal extension (~0.5–1°; *p* = 0.001) and rotation (~0.2–0.5°; *p* = 0.001–0.004), increases in hip flexion (~3°; ~65% AL−~55% PL; *p* = 0.001–0.004), internal hip rotation (~2.5°; AL + ~0–30% PL; *p* = 0.001–0.024), more knee adduction (~1°; ~80–95% AL; *p* = 0.001), and complex altered knee flexion patterns (~2–4°; AL + PL; *p* = 0.001–0.01) occurred. Complex kinematic differences between a WR and a TR were detected. This contributes to a better understanding of the incoordination in transition running.

## 1. Introduction

In a triathlon, the smooth transitioning from cycling to running is considered to be crucial for optimal performance in the final running part and the overall result [1,2,3]. The ability to quickly find one’s personal running pattern is of great importance, which contributes to optimal processes in muscular control and energy supply [1,4,5,6]. For all triathlon distances, athletes have to cope with what they colloquially describe as “brick running” (it feels like “hitting a wall of bricks”) or “running on eggs” at the beginning of the running section. This sense of incoordination, which is not found in isolated running, may result in altered running kinematics triggered by prior cycling [2,7,8].

Triathlon-specific cycling is characterized by extreme static trunk flexion and repeated hip flexion [9]. This posture is endured in order to minimize the area of wind attack, since drafting is forbidden in middle- and long-distance triathlons [10]. After the transitioning to running, an upright trunk has to be maintained and hip extension is necessary for propulsion. It is possible that these rapid changes in postural (and neuromuscular) demands cause sensorimotor perturbations [5] and, therefore, a delay in finding one’s running pattern.

The literature to date does not confirm extensive changes in the running kinematics in a transition run (TR) compared to isolated running (after a warm-up run (WR)) [4,5,11]. The studies executed to date have been conducted in laboratory settings using optical motion capture techniques, the gold standard in kinematic analyses, while the running trials have been performed on treadmills. It has been shown that after a short familiarization period, treadmill running can be similar to overground running [12,13], but (especially in the beginning) it does not realistically reflect overground running. In order to prevent falls, the subjects’ walking or running strategies are characterized by shorter step lengths, less vertical oscillation, and more anterior inclination in the trunk [14,15,16]. Since the actual kinematic differences between TR and isolated running can be expected to be rather small and predominantly at the beginning of the TR, these differences may be concealed when running on a treadmill. Investigating overground running could offer a more realistic insight in running kinematics, which can be achieved using inertial motion capture (IMC) systems, as suggested by Walsh et al. [2]. Recently, IMC systems, such as the MVN Link provided by Xsens (Xsens Technologies B.V., Enschede, The Netherlands) have become sufficiently precise. Validity studies [17,18,19,20,21], comparing IMC systems to optical motion capture (OMC) systems, have shown that for the sagittal and frontal planes, in particular, good-to-excellent data can be derived. Although the validity of these systems appears to be best for slower tasks and lower limb joints, the major causes for lower validities appear to be different depending on the applied biomechanical models [21,22].

However, only Weich et al. [3] have so far performed a transition study on a running track, analyzing data derived from inertial sensors attached to the lower limb. Their results showed acute effects in the first five minutes in both isolated running and a TR following a 30 min cycle session and, thus, no triathlon-specific running pattern; however, in a warmed-up condition, this is not considered an impairment. Furthermore, it is possible that “brick running” is characterized by changes in the running kinematics of both the lower legs and the more cranial joints due to the specific posture maintained during cycling. Overall, a better understanding of the phenomenon is crucial to further develop the transition training and to advise athletes.

To our knowledge, Weich et al. [3] (only working on the lower leg acceleration) have been the only investigators who applied a continuous data analysis method to whole step cycles. Other kinematic investigations have used single-point data analysis [23,24], for example, by comparing joint angles at the moment of the toe-off [23] or by using local Gaussian methods on time-normalized data chunks of one-minute duration [25]. Statistical methods considering whole function-like data curves are, however, crucial because analyzing gait data with single-point approaches may also result in validity and reliability issues [26]. The influence of altered conditions on continuous data can barely be expressed in a single data point. Continuous data analyses are necessary since it is unclear in which a particular moment of step cycle that a potential difference between the running pattern in isolated and transition running may occur.

In the current study, therefore, we chose to compare the starts of a WR and a TR by means of statistical parametric mapping (SPM) [27,28] applied to full step cycles. We hypothesized that the TR would be identical to the WR in all three degrees of freedom in all of the ten investigated joints (the L5S1, L4L3, L1T12, and T9T8 joints of the trunk and the ankle, knees, and hips of both legs).

## 2. Materials and Methods

### 2.1. Subjects

Sixteen healthy male triathletes (age: 32.1 ± 6.0 years; height: 1.8 ± 7.3 m; weight: 75.4 ± 7.7 kg) who trained to complete the Ironman triathlon in the then-current season sub 9 h 30 min volunteered to take part in the study. When entering the investigation, subjects reported which running and cycling intensity they were aiming and training for in their major competition of the year. Their targeted competition running velocity was 13.4 ± 0.6 km/h, and their targeted cycling intensity was 234.1 ± 18.6 watts. Subjects had performed 17.5 ± 11.7 transition training sessions in the year when they participated in the study. Subjects followed their individual training schedules.

Exclusion criteria were the intake of perception-altering substances and acute injuries or serious diseases (e.g., cardiovascular/pulmonary/renal dysfunction, neurological/psychological diseases, advanced degenerative diseases of the musculoskeletal system, and injuries that had not fully healed) that affect the quality of life or physical performance. All subjects provided written informed consent. The study was approved by the Ethics Commission of the Department of Psychology and Sports Sciences at Goethe University Frankfurt (Germany) (reference number: 2019-10).

### 2.2. Measurement System

The Xsens MVN Link system was used to record all kinematic data. The system consists of 17 miniature inertial measurement units (MTx; dimensions: 3.5 × 2.5 × 0.8 cm), which are linked via cables. Each MTx contains 3D linear accelerometers, 3D magnetometers measuring the (Earth’s) magnetic field, 3D rate gyroscopes, and a barometer. The MTx sensors, the battery (169 g; 8.5 × 6 × 2.5 cm^3^), and the “body pack” (the data lock that receives and processes data—144 g; 11.2 × 7 × 2.8 cm^3^) are mounted onto the full length Lycra suit (sizes: S–XXL) that subjects wear. The suit is provided by Xsens for optimal sensor placement and to avoid sensor movement during measurements. The suit fits close to the body without restricting movement. Sensors are mounted on Velcro straps that are attached to the arms and legs. In order to both guarantee a secure hold and safely stow the connecting cables, these areas are closed by zippers. For the hand sensors, special gloves, as well as a head band for the head sensor, are provided.

The sampling rate is 240 Hz. The data are received via WiFi by a router that is attached to the computer. The MVN Analyze software provides joint angles according to ISB (International Society of Biomechanics) standards [29]. In the current study, we included the ankle, knee, and hip joints of both legs, as well as the L5/S1, L4/L3, L1/T12, and T9/T8 back joints, all of which are provided in the software.

### 2.3. Running Track

The running track (Figure 1) was a 4 km long paved section of the track of the European Ironman Championship, which is held annually in July in Frankfurt/Main, Germany. In this event, athletes run the chosen passage four times. The course has a flat profile and only minor curves.

### 2.4. Procedure

In order to capture the subjects with similar training statuses, we attempted to schedule the recordings four-to-six weeks before their main event of the year. Therefore, all measurements were carried out between April and August 2019 in the morning (8:00–11:30 AM) on a part of the running track of the European Ironman Championships. All recordings were realized 29.1 ± 52.3 days before the main event of the individual subject. The trackers of the MVN Link IMC system were firmly attached to the Lycra suit. The feet sensors were taped on top of the athlete’s own running shoes. When the setup was complete, the calibration was performed [29]. For this procedure, the test person initially stands in a neutral position and then walks a short distance. During the subsequent processing, the orientation of the global coordinate system is set and the sensors are aligned to each other. Before the recordings, the athlete’s own triathlon racing bike was mounted onto a stationary trainer. The cycling intensity was set to the subjects’ individual aimed competition intensity (234.1 ± 18.6 watts) in the Ironman; this was controlled via a power meter during the 90 min cycling session. As security for the participants was our primary concern, cycling on open roads was not possible. The length of the cycling session on the stationary trainer was determined in consultation with various triathletes. The aim was to simulate a similar load to that which would be experienced during a competition as closely as possible, thus ensuring a realistic study design. However, the perceived effort on the trainer was significantly higher than on the open track, even at a controlled intensity, because there was no change in cycling behavior (e.g., drifting passages, curves, or changes in gradient) and no cooling winds. For these reasons, the duration of the cycling session was set to 90 min. Before the recordings, each athlete ran for 10 min at competition speed as a warm-up (no stretching or drills were performed). During the recordings, the measurement equipment (laptop, router, and the power bank) was stored in a backpack and carried by the instructor, who cycled in front of the running subject to provide a slipstream and to set the pace. Therefore, a speedometer that calculated speed via the cadence of the wheel was mounted on the bike. Running speed and cycling intensity were set to the individual’s running speed and cycling intensity in a long-distance triathlon (Figure 2). Both runs were performed once on the same day. While the WR was a 2.2 km long run, the TR was 4.2 km long because the present analysis was part of a larger project and the whole runs were separately analyzed. For the current analysis, however, we decided to only include the first 20 steps (as soon as the running speed had reached the targeted speed ± two standard deviations), since the feeling of incoordination has been reported to be strongest in the beginning of the TR and stride length and frequency are only altered directly after transitioning [30]. In order to not lose potential effects when calculating the median step cycles of both runs, we decided to only investigate the very beginnings of both runs. However, in order to guarantee for habitual running patterns, we decided to record the whole running sessions. In addition, the participants were not informed about which section was analyzed. We considered it crucial that subjects consciously prepared for a longer run. We are aware that there are acute effects even in a WR condition [3], but starting a run after a prior warm-up is the best possible condition in competition sports (rolling starts are generally not executed in competition). Moreover, the feeling of incoordination has not been reported in warm-up runs and can therefore be considered to be unique in transition running. Precisely, in the WR, a running distance of 25.21 ± 1.56 m was analyzed, and in the TR, the first 24.98 ± 2.25 m were analyzed. The exact measurement procedure is displayed in Figure 2.

### 2.5. Data Processing

Though the running course was flat, minimal changes in elevation (Figure 1) could not be avoided. Therefore, all measurements were recorded in a so-called multilevel scenario that is recommended by the manufacturer for tasks performed on uneven ground. In order to achieve the best possible data quality, the “HD reprocessing” filter was applied (MVN Analyze software). Each recording was manually inspected. Of the 22 datasets, 16 were completed. For the remaining six trials, the data curves were impaired at least in one run; these trials were removed from further analysis as the software, MVN Analyze, was not able to reconstruct the actual movement via interpolation. The high drop-out rate was due to the complex measurement procedure, by virtue of there being only one opportunity to perform a TR. The recordings were exported as .mvnx files and then compiled in the .mat format.

All further steps were performed using MATLAB R2020a. The step rate, step length, and angles of the back (L5/S1, L4/L3, L1/T12, and T9/T8), hips, knees, and ankles of both legs—each in all three degrees of freedom (abduction/adduction, internal and external rotation and flexion/extension)—were calculated. The specific spinal joints are provided in the Xsens software. The stride cycle was divided into the individual steps. Each step started with the first contact of the foot and ended with the first contact of the contralateral foot. The joint angles of the left leg were mirrored in order to be compared with the right leg. Therefore, in the subsequent analysis, a distinction between the right- and left-hand body side was no longer possible. The step cycles were further divided in active leg (AL) and a passive leg (PL) phases. The active leg phase included all time-point from the touchdown of the foot to the touchdown of the contralateral foot. Hence, it covered the stance phase plus the initial swinging phase after toe-off. The relative moment of the toe-off was at 60.4 ± 4.7% of the active leg phase. The passive leg phase included the remaining swinging phase (the foot is always in the air). For the step rate and step length, descriptive statistics were calculated.

Once the running speed had reached the mean speed by two standard deviations, the subsequent 20 steps (10 right steps and 10 left steps) were analyzed for each run. Data were time-normalized (100 points) for a vector analysis using SPM [27]. The normality of the distribution at each of the 100 time points in the 20 steps was tested using a one sample Kolmogorov–Smirnoff test in MATLAB; differences between both runs were investigated using a t-test for paired samples where data were normally distributed, while a non-parametric t-test for paired samples was applied if the data were not normally distributed via a method called statistical non-parametric mapping (SnPM) [28]. For the comparison of the mean running velocity, mean stride length, mean frequency, and the analyzed running distance between the WR and TR, a Wilcoxon matched pairs test was used when data were not normally distributed and a t-test for paired samples was applied when normal distribution occurred.

The data and the custom-written MATLAB code can be found here: https://github.com/ChristianMaurerGrubinger/Triathlon.

## 3. Results

### 3.1. Running Speed, Pacing, and Cycling Intensity

The mean individual running speed in the WR and TR were 12.77 ± 0.55 and 12.75 ± 0.69 km/h, respectively. As can be seen in Table 1, no statistically significant differences between the running conditions were found for the velocity, the stride length, and the step frequency which were calculated from the kinematic data.

### 3.2. Joint Kinematics

Since the data were not normally distributed at most time points, median running patterns and interquartile ranges (IQRs) were calculated and the non-parametric t-test for paired samples was applied to test differences between these pattern (SnPM). The analysis revealed significant differences in the trunk joints (Figure 3 and Figure 4), as well as the hip and knee in both the active and passive phases (Figure 5, Figure 6 and Figure 7). In the ankle joint, no significances were found in the sagittal plane, while in the ankle rotation and abduction, significances were very small and only covered 2–3° of the step cycle. Clinical relevance is therefore highly questionable, and the results of the ankle are not presented in detail in the current manuscript. Here, only the results of the trunk flexion and rotation, the hip flexion and rotation, and the knee flexion are presented. The SnPM, differences, and medians of the other joints can be found in the Supplemental Digital Content (SDC): Trunk Abduction (SDC 1), Hip Abduction (SDC 2), Knee Rotation and Abduction (SDC 3–4), and Ankle Flexion, Rotation, and Abduction (SDC 5–7).

### 3.3. Trunk Joints

In all four interpolated trunk joints (L5S1, L3L4, T12L1, and T8T9), the subjects showed more extension in the TR than in the WR (*p* = 0.001) at all step phases (Figure 3). In L5S1, the median difference was ~1°, while the difference was around 0.5° in each of the three more cranial joints. In addition, a statistically significant difference (*p* = 0.001–0.004) was found in the rotational axis in the flight phase (Figure 4). In the TR, the subjects showed a slight increase in the outer rotation of ~0.5° in the L5S1 joint and ~0.2° in the more cranial joints on both body sides. For the trunk, a distinction between the active and passive leg is not useful since one of both legs was active at any time to support the trunk. In Figure 3 and Figure 4, all data are included.

### 3.4. Hip

Subjects experienced significantly more hip flexion (Figure 5) in the swing phase (AL: *p* = 0.004; PL: *p* = 0.001) of the TR. This effect occurred from the toe-off until 50% of the PL phase, approximately, including the mid-swing. The median absolute difference between the runs was ~3°. Significant differences were also found in the hip rotation (Figure 6); in the TR, more internal rotation (AL: *p* = 0.001; PL: *p* = 0.002–0.024) of ~2.5° was detected in the entire AL phase and in two short sections in the PL phase, where increases of internal rotation could be observed at ~0–25% and ~90–100%.

### 3.5. Knee

The SnPM revealed a short phase (~80–95% of the AL phase) of significantly more knee adduction (*p* = 0.001) in the initial swing phase of the TR (Appendix A). The absolute median difference was ~0.5–1°. In the sagittal plane (Figure 7), more complex differences were found: at ~20–30% of the active leg phase (about mid stance), the knee was ~1.5–2° more flexed in the TR (*p* = 0.01), and at ~65–85% of the AL phase (initial swing), the knee was ~2–4° more extended in the TR. Later, in the mid-to-terminal swing (~50–80% of the PL phase), the knee was again ~4° more flexed in the TR.

## 4. Discussion

The aim of this study was to investigate potential differences in running kinematics at the beginning of a TR and a WR in elite male triathletes by using inertial sensors in overground running. This is the first study to provide kinematic data on the joints of the hip, knee, ankle, and trunk for transition running in real-world conditions. In total, the results suggested alterations in the subject’s running pattern that could have been caused by prior cycling. The comparison of the WR and TR revealed modest and absolute, but nevertheless significant, differences in the kinematics. Therefore, the hypothesis that the starts of a TR and a WR are identical has to be rejected.

This sheds new light on the matter since the literature [5,31,32] scarcely illuminates any effect of transition running on biomechanical parameters, as studies were mostly conducted in laboratory settings and time points of analysis were chosen arbitrarily instead of analyzing entire step cycles. For example, the slightly pronounced extension in the spinal joints contrasts with the findings of Hausswirth et al. [23], who found a more anteriorly inclined upper body posture using a 3D optical motion capture system to capture whole body kinematics during a treadmill run following a 60 min cycling session. This contradiction could be explained by the different conditions (treadmill vs. overground), or the study design. Though Hausswirth et al. [23] investigated joint angles at the toe-off and the touch-down, as well as the amplitude for the right leg, we analyzed the entire active and passive leg phases of both legs by means of the SnPM method in order to not miss alterations by predisposing events of potential interest.

We are aware that the absolute median differences were rather small (~0.5–1°) and the four spinal joints included were interpolated. However, these differences occurred in all four joints for all the included athletes, as well as when combining the joints to a generalized trunk movement (pelvis to sternum; this is possible when using the Xsens MVN Analyze software). Doing so added up to ~3–4° of pronounced trunk extension, which is comparable to the findings of Rendos et al. [33], who analyzed non-triathletes in a treadmill setting using a Vicon system.

To compare our results using IMU-based system with the gold standard of optical 3D motion capture systems, validation studies to compare the two systems are important. For example, the validity study by Robert-Lachaine [21] suggested that the mean root mean square error was 2.8° in all joints in faster movements. This was supported by Zhang et al. [20], who showed a great waveform similarity (a coefficient of multiple correlation of >0.96) in the sagittal plane for the leg joints in walking. To our knowledge, there has been no study validating the system in overground running, and rotational deviations (including the hip) must be interpreted with caution since the absolute differences are very small (median ~0.2–0.5°) and the rotations are not entirely precise when derived from signal processing in IMC systems [34]. This has also been a presented problem in the available validity studies [18,35,36]. Furthermore, the clinical importance of such small deviations is questionable since they may fall in the range of natural movement variation. In the following discussion, we therefore focus on the findings in sagittal plane differences located at the spine, hip, and knee.

We speculate that the crouched aero position, which is characterized by the static stretching of the back extensor muscles and extreme hip flexion, and the cyclic concentric movement of pedaling induce performance-specific fatigue in muscles. In the current study, the 90 min cycling session at competition intensity was chosen to produce the most realistic preload before the TR. The pronounced trunk extension in the TR may be a contra reaction to that preload [9]. It is possible that the spinal extensor muscles (e.g., M. erector spinae and M. multifidi) need some time to adapt the fine tuning in contraction after the long period spent in a stretched position. This is most clearly visible in the most distal joint, L5S1, but it is also evident in the more cranial back joints. One other explanation could be that after non-weight-bearing cycling when sitting on the saddle, subjects need some time to adjust their pelvis position and usual lumbar lordosis when subsequently running. Though we did not directly analyze the pelvis segment in space, the increase in hip flexion in the flight phase supports the hypothesis of altered pelvic positioning.

In general, the findings in the leg joints interestingly showed that the most prominent kinematic changes in the sagittal plane occurred in the swing phase (~3° more hip flexion, ~2–4° more knee extension at the initial swing, and ~4° more knee flexion at the mid/terminal swing). In the past, a more extended knee at the touch-down was described [31], which is not in accordance with the present findings. However, this finding came from a study executed on a treadmill after high-intensity cycling. Both factors could potentially be responsible for the contradictory findings.

Only a slight increase in the internal hip rotation appeared in the entire AL phase. Since the produced force is much greater in the stance phase (to generate the required impulse for the flight phase), it can be expected that its neuromotor activation is different from that of the swing phase. We speculate that in the rather passive swing phase during running, subjects have trouble finding a neuromotor strategy that produces their usual running pattern after having executed the cyclic movement of biking for 90 min. The pronounced hip flexion in the swing phase especially seems to resemble the hip flexion in cycling when the crank arm is on the top.

EMG research in transition running partly supports this speculation; for example, Heiden and Burnett [37] showed that vastus lateralis, vastus medialis, and rectus femoris showed different levels of activation and partial alterations in the time of activation in the flight phase. The authors suggested that the change from a concentric movement in cycling to stretch shortening cycles in running could be the reason for the altered muscle activity [37]. Though muscle activity was not investigated in the present study, such investigations should be considered for the interpretation of the results, especially since kinematic changes may be the result of altered muscle activity [38,39], among other factors. The execution of neuromotor strategies might even be affected even when kinematics are unchanged, which is what Chapman et al. [32] described for treadmill running after a 30 min cycling session (70.1 ± 13.7% change in mean EMG amplitude). The organism may alter the neural activation pattern in order to maintain the subject’s preferred movement path; this may especially be the case for highly experienced runners [40]. However, there is also evidence neglecting an influence of cycling on muscle recruitment in subsequent running [4,8,41]. The reasons for this contradicting evidence might be the differences in study protocols, but it has also been discussed that the differences in cycling conditions, despite the given intensity, might affect potential transitioning effects. For example, the cleat position in the sagittal plane might impact the gastrocnemius activity and therefore induce different levels of fatigue before running [42], but the pedaling cadence, the range of motion, the muscle lengths, the movement of the arms, and the individual positioning on the bike might also have an influence on the recruitment pattern [43,44,45]. Future studies should therefore aim to integrate EMG measurements when analyzing running patterns and bike kinematics. In the current study, the individual cycling factors were not controlled for, because in order to simulate the most realistic conditions, we encouraged our subjects to cycle in their usual manner.

While this reduction of standardization might introduce more statistical differences, it reflects the real world settings and therefore has a larger emphasis on external validity. Another limitation of using outdoor settings, however, is the difficulty to control for temperature. Our measurements were scheduled between April and August in Germany, and though we started early in the morning to ensure cool temperatures, differences could not be avoided. However, one major advantage of this prolonged data-collection period was that subjects were all approximately in the same training period shortly before or after their major event in the year (29.1 ± 52.3 days before the main event), and subjects had already performed 17.5 ± 11.7 transition training sessions in the then-current season.

In addition to the joint angles, we evaluated step length and step rate, neither of which showed any statistically significant differences between conditions. This is partly supported by the literature [24,46], although there is also evidence supporting smaller step lengths in the TR than in isolated running conditions [23,47]. However, the cycling sessions in both prior studies were shorter (20–30 min), and the intensity was reported as high [47], moderate [25], or not controlled for [23]. In the current study, the step lengths were not altered since the cycling intensity was kept low (long-distance triathlon cycling intensity).

Future studies should also consider novice triathletes [2,8,11] and athletes specialized in shorter triathlon distances, as a higher cycling intensity may have an impact on subsequent running patterns. Furthermore, there may also be gender-specific differences in transition running kinematics [48], and recruitment patterns have been shown to be different in women; for example, the gluteus medius muscle shows higher levels of activity, especially at higher intensities [49].

Based on the current findings, the phenomenon of incoordination might be better understood in the future, and “brick workouts” could be developed or adapted. Though we did not measure any neuronal parameters, the detected alterations in kinematics looked similar to that in running (e.g., increased hip flexion in the hip). We therefore suspect that the reasons for the feeling of incoordination are difficulties in adapting the neural system. When cycling, for example, the muscles generate force at a different muscle length than when running. This is controlled by reflex loops in the spinal cord and is performed over many repetitions. When suddenly changing the movement program at the central level to running in the transition run, the reflex circuits might need some time to adapt in order to regulate the corresponding muscle lengths. The brain then receives feedback via afferent pathways that coordination is disturbed.

The current methods in transition training, which our athletes had regularly performed in their then-current season, were apparently not entirely adequate to eliminate the changes in the kinematic pattern. These demonstrated alterations were not limited to a single joint, as they were also found in the spine, hip, and knee in different movement planes at different phases of the step cycle. This complex shift may be a reason why athletes fail to more precisely describe their incoordination beyond “brick running” or “running on eggs.” However, based on the current findings, we cannot conclude whether these changes lead to a decrease in running performance or running economy, which has been described before [31]. If this is confirmed by future studies in outdoor settings, one possible approach might be to include plyometric training, which seems to be beneficial even though its effects have not yet been compared to traditional brick training, in the training process [50]. Another possibility might be to include short stretching sessions or foam rolling for the back and hip in the transition period, which have occasionally been used by individual athletes. However, the loss of time caused by either the stretching or the transitioning effect will have to be carefully evaluated.

## 5. Conclusions

Comparing the first 20 steps of transition and isolated running in overground conditions revealed differences in the running pattern of elite male triathletes. In the transition run, subjects were found to run with a pronounced trunk extension, increased hip flexion in the flight phase, more knee extension in the initial swing, and increased flexion in the terminal swing. This indicates that prior cycling induces difficulties in finding one’s preferred running pattern, supporting the anecdotal reports of incoordination at the beginning of a transition run in triathlons. Current methods in transition training need to be adapted in order to produce smooth transitions from bike to running.

## Figures and Tables

**Figure 1 sensors-21-04869-f001:**
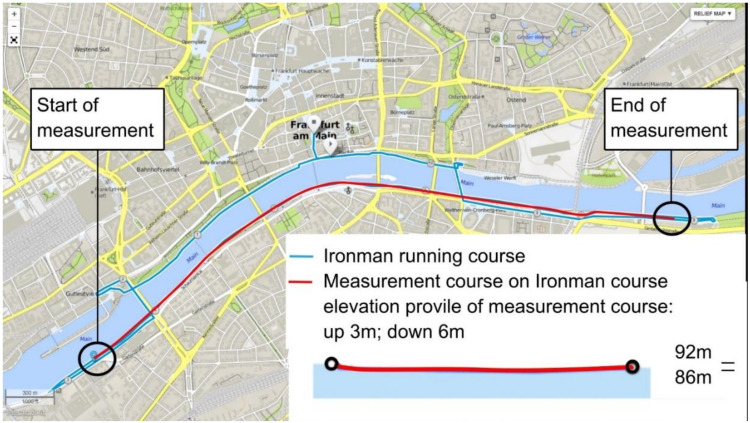
Running track—a part of the Ironman European Championship in Frankfurt/Main, Germany. The running track of the competition is marked in red, including the kilometer marks. In the competition, the course has to be completed four times. The running track used in the study is marked in red (4 km).

**Figure 2 sensors-21-04869-f002:**
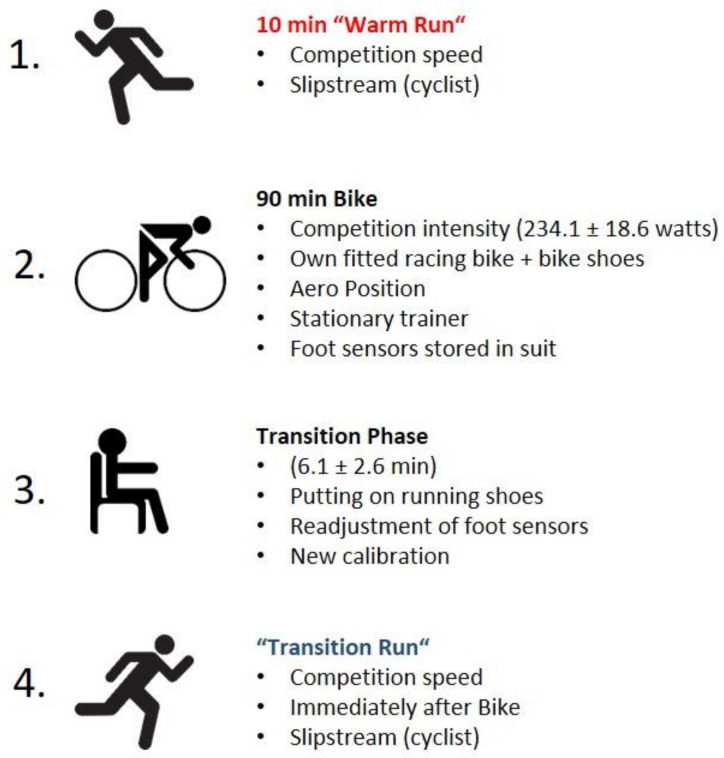
Procedure: both runs were performed once on the same day. Subjects were allowed to drink or eat ordinary sports nutrition during the cycling session. The new calibration during the transition phase was necessary because the motion trackers had to be re-applied to the running shoes. Running speed was set to the individual’s running speed in a long-distance triathlon (13.37 ± 0.6 km/h).

**Figure 3 sensors-21-04869-f003:**
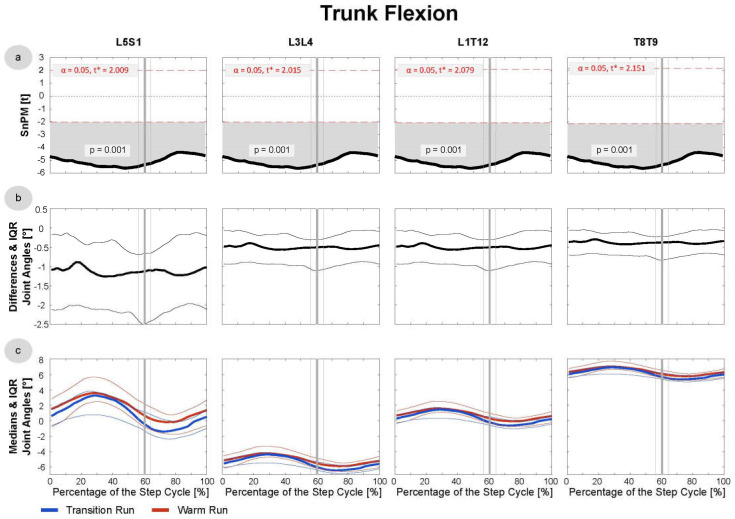
Trunk Flexion/Extension: data were derived from the first 20 steps of the TR and WR of all 16 included subjects. (**a**) SnPM: in cases of statistical significance, the black curve lies outside the red line. (**b**) Median differences and the IQR: excursions above the 0-axis indicate more flexion in the TR/extension in the WR; excursions below the 0-axis indicate the opposite. (**c**) The medians and the IQR of the transition and warm-up runs are shown in blue and red, respectively. The vertical bold line (grey) indicates the mean toe-off at 60.4% of the active leg phase, and the thin lines indicate the SD of 4.7%.

**Figure 4 sensors-21-04869-f004:**
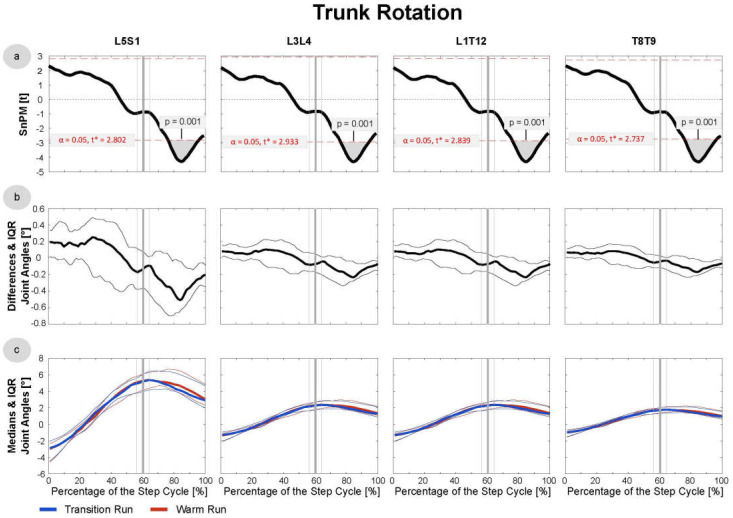
Trunk Rotation: data were derived from the first 20 steps of the TR and WR of all 16 included subjects. (**a**) SnPM: in cases of statistical significance, the black curve lies outside the red line. (**b**) Median differences: with IQR, excursions above the 0-axis indicate more internal rotation in the TR/external rotation in the WR; excursions below the 0-axis indicate the opposite. (**c**) The medians and the IQR of the transition and warm-up runs are shown in blue and red, respectively. The vertical bold line (grey) indicates the mean toe-off at 60.4% of the active leg phase, and the thin lines indicate the SD of 4.7%.

**Figure 5 sensors-21-04869-f005:**
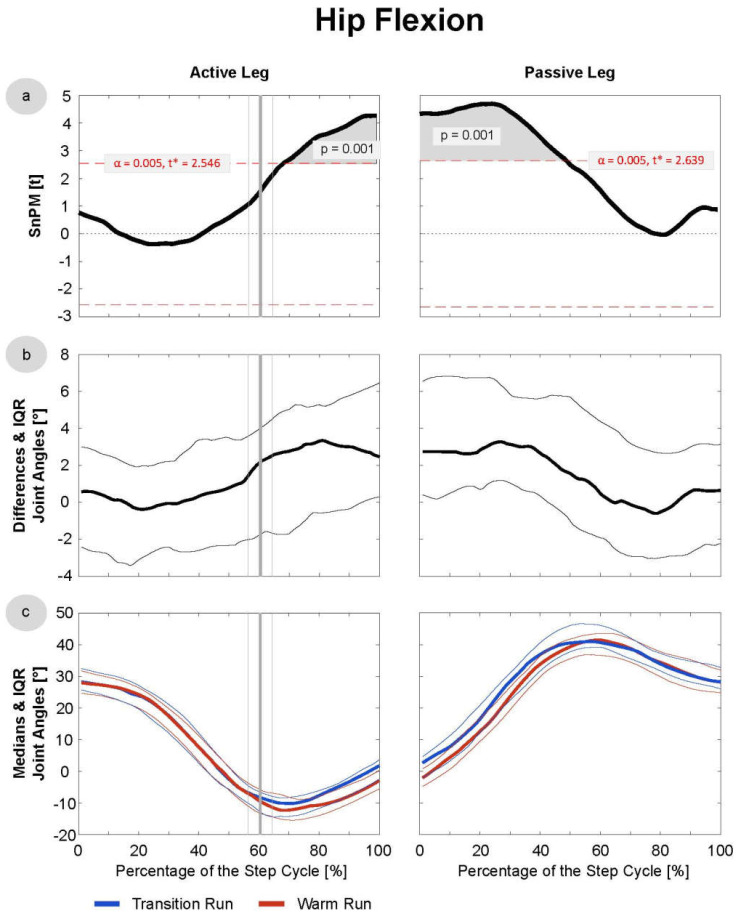
Hip Flexion/Extension: data were derived of the first 20 steps of the TR and WR of all 16 included subjects. Active and passive leg joints include right and left leg data, and the latter were mirrored for that purpose. (**a**) SnPM: in cases of statistical significance, the black curve lies outside the red line. (**b**) Median differences: with IQR, excursions above the 0-axis indicate more flexion in the TR/extension in the WR; excursions below the 0-axis indicate the opposite. (**c**) The medians with the IQR of the transition and warm-up run are shown in blue and red, respectively. The vertical bold line (grey) indicates the mean toe-off at 60.4% of the active leg phase, and the thin lines indicate the SD of 4.7%.

**Figure 6 sensors-21-04869-f006:**
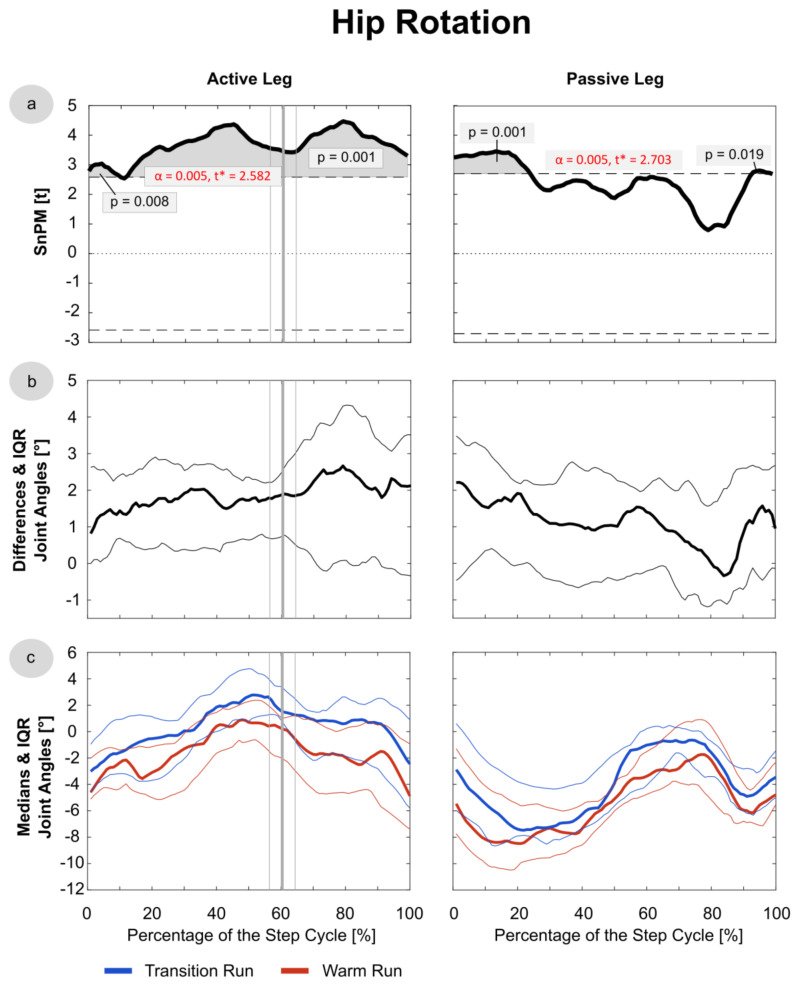
Hip Rotation: data were derived of the first 20 steps of the TR and WR of all 16 included subjects. Active and passive leg joints include right and left leg data, and the latter were mirrored for that purpose. (**a**) SnPM: in cases of statistical significance, the black curve lies outside the red line. (**b**) Median differences: with IQR, excursions above the 0-axis indicate more internal rotation in the TR/external rotation in the WR; excursions below the 0-axis indicate the opposite. (**c**) The medians with IQR of the transition and warm-up run are shown in blue and red, respectively. The vertical bold line (grey) indicates the mean toe-off at 60.4% of the active leg phase, and the thin lines indicate the SD of 4.7%.

**Figure 7 sensors-21-04869-f007:**
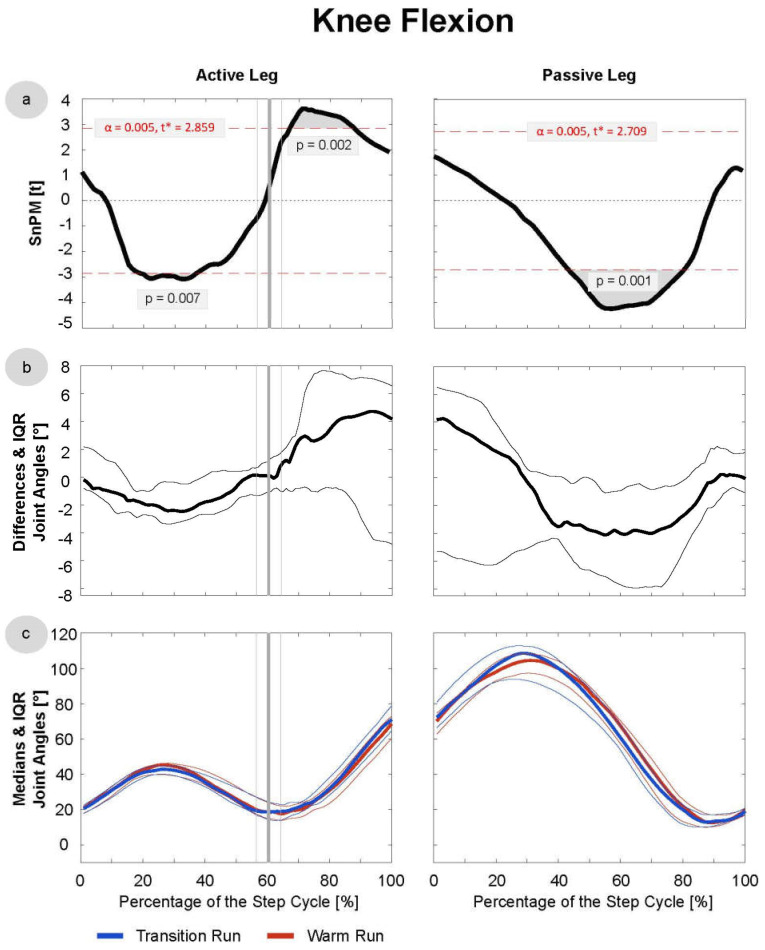
Knee Flexion/Extension: data were derived of the first 20 steps of the TR and WR of all 16 included subjects. Active and passive leg joints include right and left leg data, and the latter were mirrored for that purpose. (**a**) SnPM: in cases of statistical significance, the black curve lies outside the red line. (**b**) Median differences: with IQR, excursions above the 0-axis indicate more flexion in the TR/extension in the WR; excursions below the 0-axis indicate the opposite. (**c**) Medians with IQR of the transition and warm-up run are shown in blue and red, respectively. The vertical bold line (grey) indicates the mean toe-off at 60.4% of the active leg phase, and the thin lines indicate the SD of 4.7%.

**Table 1 sensors-21-04869-t001:** Velocity, stride length, step frequency, and distance of both analyzed runs. The analyzed running distances were different, since the first 10 steps of each leg were included for each athlete.

	Mean Velocity (km/h)	Mean Stride Length (m)	Mean Frequency (Steps Per Minute)	Analyzed Running Distance
Warm-Up Run	12.77 ± 0.55	2.52 ± 0.16	84.63 ± 17.19	25.21 ± 1.56
Transition Run	12.75 ± 0.69	2.50 ± 0.22	85.36 ± 19.82	24.98 ± 2.25
*p*-value	0.91	0.53	0.07	0.46

## Data Availability

The data and the custom-written MATLAB code can be found here: https://github.com/ChristianMaurerGrubinger/Triathlon.

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
