# Peer review of "Comparison of Joint Kinematics in Transition Running and Isolated Running in Elite Triathletes in Overground Conditions"

_sensors, 2021, doi:10.3390/s21144869_

Round 1
Reviewer 1 Report
Triathletes, therapists and coaches alike should all find this research article on injury prevention and performance enhancement very interesting. Having said that however, the authors should more clearly explain the following points.
Is WR “run after warm-up (line 10)” or “warm run" (figure 2)?
Why did the authors report the spinal segments individually? The reason should be explained. Furthermore, movement of L5 to Th8 were determined. Why were Th10 and Th11 not measured? Please explain. It would also be interesting to know the ROM at trunk flexion and rotation as well.
The authors hypothesis that the starts of a TR and a WR were identical was not supported in the present study. However, they only analyzed the first 20 steps. I support the thought that the prior movement affects the subsequent movement. However, how about analyzing a longer time period for differences? Lastly, please state the relevance and implication(s) of the results at the end of the discussion.
As for hip rotation (Fig.6) and knee flexion (Fig.7), why was the SD so large after the 60 % phase? Could measurement errors explain this? Perhaps the authors provide an explanation?
Line 87: 1.83 should be 1.8 (significant digit)
Line 129: Please explain the Lycra suit and how and where the system was attached more clearly.
Line 167-170: The explanation of Fig. 2 should be in the Methods section.
Line 212-214: The explanation of Table 1 should be in the Methods section.
Line 221: Fig. 3+4 should be Fig. 3 and 4.
Line 395-397: Stretching might affect the movement so please determine the warm-up before the measurement.
Author Response
Response to Reviewer 1:
Triathletes, therapists and coaches alike should all find this research article on injury prevention and performance enhancement very interesting. Having said that however, the authors should more clearly explain the following points.
à Thank you very much for your time and effort in reviewing our manuscript. We believe your comments and recommendations enhanced the quality of this paper. Please find our comments on your specific questions below.
Is WR “run after warm-up (line 10)” or “warm run" (figure 2)?
à Thank you for mentioning, we have adapted figure 2.
Why did the authors report the spinal segments individually? The reason should be explained. Furthermore, movement of L5 to Th8 were determined. Why were Th10 and Th11 not measured? Please explain. It would also be interesting to know the ROM at trunk flexion and rotation as well.
à We included the named trunk joints specifically, because these are the provided joints in the Xsens software. This allows for a more detailed insight in where exactly in the spine the changes occurred. However, if one is interested in the trunk flexion (pelvis versus sternum), the flexion angles of the named joints can simply be added. We changed the manuscript in the Methods section (Line 242ff): “All further steps were performed using Matlab R2020a. The step rate, step length and angles of the back (L5/S1; L4/L3; L1/T12; T9/T8), hips, knees and ankles of both legs, each in all three degrees of freedom (abduction/adduction, internal and external rotation and flexion/extension), were calculated. The specific spinal joints are provided in the Xsens software.”
In the discussion as well, we refer to that (Line 387ff): “We are aware that the absolute median differences are rather small (~0.5-1°) and the four spinal joints included are interpolated. However, these differences occurred in all four joints, for all the included athletes and also when combining the joints to a generalized trunk movement (pelvis to sternum; this is possible when using the Xsens software MVN Analyze). Doing so, adds up to ~3-4° of pronounced trunk extension which is comparable to the findings of Rendos et al. (33) who analyzed non-triathletes in a treadmill setting using a Vicon system.”
The authors hypothesis that the starts of a TR and a WR were identical was not supported in the present study. However, they only analyzed the first 20 steps. I support the thought that the prior movement affects the subsequent movement. However, how about analyzing a longer time period for differences? Lastly, please state the relevance and implication(s) of the results at the end of the discussion.
à Thanks for this comment. We are aware, that the first 20 steps of each run are a very short running distance. However, the athletes reported, that the feeling of incoordination is present mostly only at the very beginning of the transition run. As it is unclear, for how many steps or for which distance this feeling is perceived in different athletes, we decided to analyze only the very beginning. As we have described in the “procedure” section (Line 213ff): “In order, not to lose potential effects when calculating the median step cycles of both runs, we decided to investigate only the very beginnings of both runs. Analyzing the start of in fact longer runs, is also beneficial, since the instruction of only running 20 steps would hardly have offered realistic running performances. We considered it crucial, that subjects consciously prepared for a longer run. We are aware, that there are acute effects even in a WR condition (3), but starting a run after a prior warm-up is the best possible condition there is in competition sports (rolling starts are not executed). Moreover, the feeling of incoordination has not been reported in warm runs and can therefore be considered to be unique in transition running.”
As we have recorded much longer running parts, we also intend to analyze in what way the running pattern develops over longer periods in both runs in future publications. But this however, would not fit in the current article.
We also agree, that in the original version of the manuscript the end of the discussion section could be improved. In the revised version, the last page of the discussion covers training implications and the relevance of the current findings (Line 478ff): “Based on the current findings, the phenomenon might be better understood in future and “brick workouts” could be developed or adapted. Although we have not measured any neuronal parameters, the detected alterations in kinematics look slightly similar to that in running (eg. increased hip flexion in the hip). We, therefore, suspect that the reasons for the feeling of incoordination are difficulties in adapting the neural system. When cycling, for example, the muscles generate force at a different muscle length than when running. This is controlled by reflex loops in the spinal cord and is performed over many 100 repetitions. When suddenly changing the movement program at the central level to running in the Transition Run, the reflex circuits might need some time of adaptation to regulate the corresponding muscle lengths. The brain then receives feedback via afferent pathways that coordination is disturbed.
Apparently, the current methods in transition training, which our athletes had regularly performed in the concurrent season, are not entirely adequate to eliminate the changes in the kinematic pattern. These demonstrated alterations were not limited to a single joint but were also found in the spine, hip and knee in different movement planes at different phases of the step cycle. This complex shift may be a reason why athletes fail to describe the incoordination more precisely, other than “brick running” or “running on eggs”. However, based on the current findings we cannot conclude whether these changes lead to a decrease in running performance or running economy, which has been described before (31). If this should be confirmed by future studies in outdoor settings, one possible approach might be to include plyometric training in the training process, which seems to be beneficial, but its effect has not yet been compared to traditional brick trainings (50). Another possibility might be to include short stretching sessions or foam rolling for the back and hip in the transition period, which has occasionally been reported for individual athletes. However, the loss of time of either the stretching or caused by the transitioning effect will have to be evaluated carefully. “
As for hip rotation (Fig.6) and knee flexion (Fig.7), why was the SD so large after the 60 % phase? Could measurement errors explain this? Perhaps the authors provide an explanation?
à The 60% phase is the stance phase, after that, the subjects lift the foot from the ground. We speculate, that the larger SDs are due to larger differences in the movement pattern in this flight phase. Also, this might reflect the individually different running pattern between athletes. Measurement errors are rather unlikely, because in this case, the SD would have been larger in the entire step cycle. The trackers are firmly attached to the suit and the body, so a measurement error can rather not be expected after toe off. If the sensors were deteriorated by for example wobbling mass, the SD would have been larger after touch down.
Line 87: 1.83 should be 1.8 (significant digit)
à Thanks for noticing, we changed it to 1.8 (Line134)
Line 129: Please explain the Lycra suit and how and where the system was attached more clearly.
à We have expanded on this in Line 157ff: “The suit is provided by Xsens for optimal sensor placement and to avoid sensor movement during measurements. the suit fits close to the body, but without restricting movement. Velcro straps are attached along the arms and legs, on which the sensors are mounted. in order to guarantee both a secure hold and to stow the connecting cables safely, these areas are closed by zippers. For the hand sensors, special gloves are provided, as well as a head band for the head sensor.”
Line 167-170: The explanation of Fig. 2 should be in the Methods section.
à Might there be a misunderstanding? Fig. 2 as well as the caption and the link in the text are in the Procedure section, which is part of the Methods section. Please let us know in case we did not interpret your comment correctly.
Line 212-214: The explanation of Table 1 should be in the Methods section.
à Thank you for the comment. We moved the information on the statistics to the Method section (statistical analysis), and the analyzed running distance was moved there as well (paragraph on procedure). However, we prefer to keep the Table in the results section as the measured stride lengths and frequencies are part of the kinematic results.
Line 221: Fig. 3+4 should be Fig. 3 and 4.
à Thank you for noticing, we changed it (Line 285).
Line 395-397: Stretching might affect the movement so please determine the warm-up before the measurement.
à We expanded on this in the procedure section (Line 201f): “Before the recordings, each athlete ran for 10min at competition speed as a warm-up (no stretching or drills were performed).”

Reviewer 2 Report
The authors attempted to analyze the joint kinematics in transition running and isolated running in elite triathletes in overground conditions. All in all, I hope the authors can restate the significance of the present work in the manuscript. I recommend a couple of major changes. Please see my specific comments below.
Specific comments:
Introduction:
1.line 25, revise “to be” into “as”.
2.line 27-28, the sentence is incomprehensible
3.line 29-30 What are “bricking running” and “running on eggs”? Give more detail describe the definition.
4.line 34, the sentences “This posture is endured in order to minimize the frontal plane and “, delete and. Please clearly show what the joint will minimize in the frontal plane.
Materials and Methods
- line 190-191 The angles were assigned to an active leg(AL) and a passive leg(PL). Which leg is the active leg and passive leg? It could be clearer.
- What the running speed of the beginning of a transition run and a warm run?
- Why collect the data from April to August? Different times collect the data how to control the data error?
Results:
- In the results section, the authors analyzed only the trunk, hip and knee. How was the joint of the ankle affected?
- There was no test data related to the picture (results of SPM analyse) shown in the manuscript.
- The pictures need to be reformatted and edited to make them more aesthetically pleasing and accessible to the reader.
Discussion:
- If authors, in the discussion, compare their results with results obtained by other scientists and find some differences, they should try to explain the reason for these differences.
- The seventh paragraph of the discussion is not related to the results of your research, and please rewrite it. Some recently studies could be referenced in the discussion:
Development and Preliminary Evaluation of a Lower Body Exosuit to Support Ankle Dorsiflexion. Appl. Sci. 2021, 11, 5007. https://doi.org/10.3390/app11115007
Gender Differences in Kinematic Analysis of the Lower Limbs during the Chasse Step in Table Tennis Athletes. Healthcare 2021, 9, 703. https://doi.org/10.3390/healthcare9060703
Sex and Limb Differences in Lower Extremity Alignment and Kinematics during Drop Vertical Jumps. Int. J. Environ. Res. Public Health 2021, 18, 3748. https://doi.org/10.3390/ijerph18073748
Relationship Between Isometric Hip Torque With Three Kinematic Tests in Soccer Players. Physical Activity and Health, 4(1), 142–149. DOI: http://doi.org/10.5334/paah.65
- The author think the kinematic differences contribute to a better understanding of the incoordination in transition running. I agree author’s views, but I also believe that it would be better if the author discuss this more.
Author Response
Response to Reviewer 2:
The authors attempted to analyze the joint kinematics in transition running and isolated running in elite triathletes in overground conditions. All in all, I hope the authors can restate the significance of the present work in the manuscript. I recommend a couple of major changes. Please see my specific comments below.
à Thank you very much for your effort and the helpful recommendations on our manuscript. We are sure, this improved the quality of the article. We tried to include all recommendations and rewrote several parts. Please see the specific comments. We suspect, that there was a shift in the line numbering. We nevertheless tried our best to find your comments and we believe, we found all of them. If we did miss something or if there was a misunderstanding, please feel free to let us know.
Specific comments:
Introduction:
1.line 25, revise “to be” into “as”.
à Done (line 95).
2.line 27-28, the sentence is incomprehensible
à Thanks for mentioning. We rephrased for clarity in line 74ff:“ The ability to quickly find one's personal running pattern is of great importance, since here the energy supply and muscular control can be expected to be optimal (1, 4-6).”
3.line 29-30 What are “bricking running” and “running on eggs”? Give more detail describe the definition.
à “Running on eggs”, but especially “brick running” are common phrases in triathlon to describe the phenomenon of incoordination after transitioning from bike to running. We have tried to rephrase this for more clarity (Line 76ff): “For all triathlon distances, athletes have to cope with what they describe colloquially as “brick running” (it feels like “hitting a wall of bricks”) or “running on eggs” at the beginning of the running section”
4.line 34, the sentences “This posture is endured in order to minimize the frontal plane and “, delete and. Please clearly show what the joint will minimize in the frontal plane.
à Thanks for noticing (Line 81f). We removed the “and” and inserted it before “therefore”: “Triathlon specific cycling is characterized by extreme static trunk flexion and repeated hip flexion (9).
Concerning your second question about the frontal plane, we believe this is a misunderstanding. We rephrased for clarity (Line 82f): “This posture is endured in order to minimize the area of wind attack, since drafting is forbidden in middle and long-distance triathlon (10).”
Materials and Methods
line 190-191 The angles were assigned to an active leg(AL) and a passive leg(PL). Which leg is the active leg and passive leg? It could be clearer.
à Thank you for mentioning. Do you mean line 230-239? We have tried to rewrite this section for clarity: Line 246ff “Each step started with the first contact of the foot and ended with the first contact of the contralateral foot. The joint angles of the left leg were mirrored in order to be compared with the right leg. Therefore, in the subsequent analysis, a distinction between the right- and left-hand body side is not possible anymore. The step cycles were further divided in active leg (AL) and a passive leg (PL) phases. The active leg phase includes all time-point from the touchdown of the foot until the touchdown of the contralateral foot. Hence, it covers the stance phase plus the initial swinging phase after toe-off. The relative moment of the toe-off was at 60.4 ± 4.7 % of the active leg phase. The passive leg includes the remaining swinging phase (the foot is always in the air).”
What the running speed of the beginning of a transition run and a warm run?
à See line 256f: “Once the running speed had reached the mean speed by two standard deviations, the subsequent 20 steps (10 right steps and 10 left steps) were analyzed for each run.”
To further clarify this question we have also included this in line 210f: “(as soon as the running speed had reached the targeted speed ± two standard deviations)”
Why collect the data from April to August? Different times collect the data how to control the data error?
à In order to capture the subjects in similar training status, we attempted to schedule the recordings four to six weeks before their main event of the year. As the main triathlon season goes from spring until August/September, this was the time window for our recordings. We have added this in line 181ff: In order to capture the subjects in similar training status, we attempted to schedule the recordings four to six weeks before their main event of the year. Therefore, all measurements were carried out between April and August 2019 in the morning (8:00-11:30 AM) on a part of the running track of the European Ironman Championships. All recordings were realized 29.1 ± 52.3 days before the main event of the individual subject.“ As each subject was individually equipped with the measurement system and the software version was kept the same, we did not expect any data error due to that time window.
Results:
In the results section, the authors analyzed only the trunk, hip and knee. How was the joint of the ankle affected?
à Thank you for mentioning, we have tried to clarify this in more detail (Line 286ff): “In the ankle joint, in the sagittal plane no significances occurred, while in the ankle rotation and abduction significances were very small and covered only 2 – 3° of the step cycle. Clinical relevance, therefore, is highly questionable, and the results of the ankle are not presented in detail in the current manuscript.”
There was no test data related to the picture (results of SPM analyse) shown in the manuscript.
à Thank you for the question. Indeed, all test data (SnPM t) is visualized in the figures. The curves of the a) parts of the figures indicate the p-value for each time-point analyzed. For active leg and for passive leg phases respectively, 100 time points were analyzed. In fact, Pataky, the inventor of this method himself, suggests to show the results as a plot in his examples in matlab (https://spm1d.org/DocumentationMatlab.html). As this large amount of data can hardly be shown in a table, we have decided to use the graphics as well. The most important results (degrees, phases of the step cycles and p-values) are described in the results section.
The pictures need to be reformatted and edited to make them more aesthetically pleasing and accessible to the reader.
à In the figures we tried to graphically display the complex results as obvious and as clear as possible, and in our opinion, this is the most suitable way. We are not sure, what exactly should be improved, but we welcome every clear proposal and will try to include them. Of course, in the word document, the figures are not displayed in the best quality. In the zip folders, all graphics are included in the best quality, does this maybe solve the problem?
Discussion:
If authors, in the discussion, compare their results with results obtained by other scientists and find some differences, they should try to explain the reason for these differences.
à We have shown significant differences in the running pattern between an isolated and a transition run, which is partly contradicting other authors publications. Many studies haven’t found kinematic differences, or if they found some, they are not in accordance with our findings. However, it is hard to speculate why exactly the differences occurred. In several paragraphs we have emphasized, that the study designs differ largely. Other authors had their subjects run on treadmills, which especially at the beginning of a run can cause trouble in finding one’s personal running pattern. Potentially, this masked a transition running effect. Also the cycling intensity of other authors was mostly shorter. Also, other authors analyzed only distinct moments in the step cycle, like the toe off and the touch down. Potentially, they have missed changes in kinematics, since in our study, the major finding occurred in the swing phase.
The seventh paragraph of the discussion is not related to the results of your research, and please rewrite it. Some recently studies could be referenced in the discussion:
Development and Preliminary Evaluation of a Lower Body Exosuit to Support Ankle Dorsiflexion. Appl. Sci. 2021, 11, 5007. https://doi.org/10.3390/app11115007
Gender Differences in Kinematic Analysis of the Lower Limbs during the Chasse Step in Table Tennis Athletes. Healthcare 2021, 9, 703. https://doi.org/10.3390/healthcare9060703
Sex and Limb Differences in Lower Extremity Alignment and Kinematics during Drop Vertical Jumps. Int. J. Environ. Res. Public Health 2021, 18, 3748. https://doi.org/10.3390/ijerph18073748
Relationship Between Isometric Hip Torque With Three Kinematic Tests in Soccer Players. Physical Activity and Health, 4(1), 142–149. DOI: http://doi.org/10.5334/paah.65
à Are you referring to the paragraph about the EMG studies? Of course, we did not measure muscle activity in our study, however, we are still convinced that this paragraph serves well as further potential explanation on how and why the kinematic changes occur. In the past we ourselves have gathered some experience in the EMG measurements of cycling, as well as running, we have now included some references in the manuscript. But we mainly focus on citing studies, clearly referring to running or triathlon. in order to clarify this in the manuscript we have included some explanations (Line 432ff): “EMG research in transition running partly supports this speculation; for example, Heiden and Burnett (37) have shown that vastus lateralis, vastus medialis, rectus femoris showed different levels of activation and partly alterations in the time of activation in flight phase. The authors suggest, that the change from a concentric movement in cycling to stretch shortening cycles in running could be the reason for the altered muscle activity (37). Although muscle activity was not investigated in the present study, such investigations should be considered for the interpretation of the results, especially since kinematic changes may be the result of altered muscle activity (38, 39), among other factors. The execution of neuromotor strategies might even be affected even when kinematics are unchanged, which is what Chapman et al. (32) described for treadmill running after a 30 min cycling session (70.1 ± 13.7% change in mean EMG amplitude)…”
The studies you proposed instead are much less related to the cyclic movement of running. We do not see, in how far an investigation on an acyclical movement like the chasse step in table tennis can help explain or interpret our current findings. Also, an exosuit for ankle dorsiflexion for rehabilitation scarcely helps in analyzing running pattern in healthy highly trained athletes. But we agree, that the kinematics discovered in male athlete are not uniformly transferable to women. We have included your reference by Chun et al. (Line 474ff): “Furthermore, there may also be gender-specific differences in transition running as kinematics (49) and recruitment patterns have been shown to be different in women; here for example, the gluteus medius muscle shows higher activities, especially at higher intensities (50).”
The author think the kinematic differences contribute to a better understanding of the incoordination in transition running. I agree author’s views, but I also believe that it would be better if the author discuss this more.
à Thank you for the advice. We also agree, that in the original version of the manuscript the end of the discussion section could be improved. In the revised version, the last page of the discussion covers training implications and the relevance of the current findings (Line 478ff): “Based on the current findings, the phenomenon might be better understood in future and “brick workouts” could be developed or adapted. Although we have not measured any neuronal parameters, the detected alterations in kinematics look slightly similar to that in running (eg. increased hip flexion in the hip). We, therefore, suspect that the reasons for the feeling of incoordination are difficulties in adapting the neural system. When cycling, for example, the muscles generate force at a different muscle length than when running. This is controlled by reflex loops in the spinal cord and is performed over many 100 repetitions. When suddenly changing the movement program at the central level to running in the Transition Run, the reflex circuits might need some time of adaptation to regulate the corresponding muscle lengths. The brain then receives feedback via afferent pathways that coordination is disturbed.
Apparently, the current methods in transition training, which our athletes had regularly performed in the concurrent season, are not entirely adequate to eliminate the changes in the kinematic pattern. These demonstrated alterations were not limited to a single joint but were also found in the spine, hip and knee in different movement planes at different phases of the step cycle. This complex shift may be a reason why athletes fail to describe the incoordination more precisely, other than “brick running” or “running on eggs”. However, based on the current findings we cannot conclude whether these changes lead to a decrease in running performance or running economy, which has been described before (31). If this should be confirmed by future studies in outdoor settings, one possible approach might be to include plyometric training in the training process, which seems to be beneficial, but its effect has not yet been compared to traditional brick trainings (50). Another possibility might be to include short stretching sessions or foam rolling for the back and hip in the transition period, which has occasionally been reported for individual athletes. However, the loss of time of either the stretching or caused by the transitioning effect will have to be evaluated carefully. “

Round 2
Reviewer 1 Report
The authors have thoroughly considered the suggestions and criticisms from the reviewers and revised the manuscript accordingly.